# An evolutionary algorithm based on approximation method and related techniques for solving bilevel programming problems

**Yuhui Liu**[1]*, **Hecheng Li**[2]*, **Huafei Chen**[3], **Mei Ma**[2]

**1** School of Mathematics and Physics, Qinghai University, Xining, China, **2** School of Mathematics and Statistics, Qinghai Normal University, Xining, China, **3** School of Mathematics and Statistics, Sichuan University of Science and Engineering, Zigong, China

☯ These authors contributed equally to this work.
\* liuyuhui_02@163.com (YL); lihecheng@qhnu.edu.cn (HL)

**Data Availability Statement:** The minimal dataset is located at https://doi.org/10.5061/dryad. wh70rxwr1.

## Abstract

In the engineering and economic management fields, optimisation models frequently involve different decision-making levels. These are known as multi-level optimisation problems. Because the decision-making process of such problems are hierarchical, they are also called a hierarchical optimisation problems. When the problem involves only two-level decision-making, the corresponding optimisation model is referred to as a bilevel programming problem(BLPP). To address the complex nonlinear bilevel programming problem, in this study, we design an evolutionary algorithm embedded with a surrogate model-that it is a approximation method and correlation coefficients. First, the isodata method is used to group the initial population, and the correlation coefficients of the individuals in each group are determined based on the rank of the leader and follower objective functions. Second, for the offspring individuals produced by the evolutionary operator, the surrogate model is used to approximate the solution of the follower's programming problem, during which the points in the population are screened by combining the correlation coefficients. Finally, a new crossover operator is designed by the spherical search method, which diversifies the generated offspring. The simulation experimental results demonstrate that the proposed algorithm can effectively obtain an optimal solution.

## Introduction

### Problem models

BLPP is a typical representative of multilevel hierarchical optimisation problems. In contrast to multi-objective optimisation, the decision makers in BLPP are at two different levels. This hierarchical structure often leads to a corresponding problem that is neither convex nor differentiable, which is also strongly non-deterministic polynomial hard as well. The BLPP model is

**Funding:** The National Natural Science Foundation of China, 61966030, Hecheng Li The Program of Science and Technology International Cooperation Project of Qinghai Province, 2022-HZ-806. The Youth Foundation of Qinghai Normal University, 2018zr001.

**Competing interests:** The authors have declared that no competing interests exist.

expressed as follows:

$$\begin{cases} \min_{x} F(x, y) \\ s.t. G(x, y) \leq 0 \\ \min_{y} f(x, y) \\ s.t. g(x, y) \leq 0 \end{cases} \tag{1}$$

Here, $x = (x_1, \ldots, x_n)$ and $y = (y_1, \ldots, y_m)$ are the leader's and follower's variables. $F: R^{n+m} \rightarrow R$ and $f: R^{n+m} \rightarrow R$ are the leader's and follower's objective function. $G: R^{n+m} \rightarrow R$ and $g: R^{n+m} \rightarrow R$ are the leader's and follower's constraints, respectively.

It can be seen from (1) that BLPP is an interaction optimisation model between two decision makers with their own objectives. The decision-making procedure is executed as follows: The leader, located at upper level, makes a decision by selecting a variable value. Then the follower observes the follower's selection and responds to the leader's decision by optimizing his/her objective and providing an optimal solution $y$. The point pair $(x, y)$ is called a feasible point of BLPP. The bilevel optimisation arms to select a value $x$ that optimizes the leader's objective among all feasible points.

During the optimisation procedure, the leader influences the follower by providing a value $x$ fixed as a parametric value. Subsequently, the follower alters the values of the objective and constraints at the leader's problems by reacting a value $y$ to address the leader's problem. When the optimal solution $(x, y)$ is obtained, the optimisation process stops.

The nested nature of BLPP poses a number of additional challenges compared with conventional single-level optimization problems. In particular:

1. The leader's behaviour of a bilevel problem may be nonlinear, even if the problems at both levels are linear. It has been proved that BLPP are non-deterministic polynomial hard problem.

2. Theoretically, a leader's solution is considered valid/feasible only if the corresponding follower's variables are the true global optimum of the follower problem. Global optimality can only be assured in very limited cases, such as convex and linear problems. However, for most nonlinear and black-box problems, it is not possible to ensure global optimality.

3. In the deceptive case, incorrect follower's optimal values may cause the objective value to be better than the leader's true optimal value, which poses a severe challenge to the ranking strategy used in evolutionary optimization technology.

4. Due to each variable of the leader needs to solve the follower programming problem to obtain a feasible solution of the BLPP, the cost of calculation is significantly high.

In particularly, if the problem is mathematically not suited to be solved using exact techniques, thus, evolutionary/hybrid techniques are used.

## Related work

With the advancements and developments in human science and technology, various increasingly complex optimisation models have emerged, particularly in the fields of equipment installation, task scheduling, production planning, line optimisation, software design, and tariff setting. Therefore, how these models can be effectively solved has emerged as an important main research topic in the optimisation field. Over the past few decades, many optimisation

research results have mainly focused on single-objective and multi-objective optimisation models, whereas relatively few studies have been conducted on hierarchical optimisation models.

Leveraging the one-time decision theory to produce multiple short-life cycle products, Zhu and Guo [1]studied BLPP applications with follower problem operators for manufacturers and used classical optimisation methods to solve them. *Nasrolahpour* et al. [2] developed an energy storage system for merchant pricing based on a two-tier complementary model that can determine the most favourable transaction behaviour. Under a multi-objective framework, Ahmad et al. [3] proposed a simple multi-objective bilevel linear programming method, which considered reservoir managers and multiple water-use departments as a hierarchical structure to optimally allocate limited water resources. More applications of optimisation models can be found in the literature [4–13].

Many practical applications have promoted theoretical research on bilevel programming problems, such as developing efficient algorithms and obtaining optimality conditions. However, owing to the computational complexity of the BLPP itself, it is often very challenging to adopt traditional optimisation methods based on gradient information to address such problems. Currently, only special bilevel programming approachs, such as linear and convex quadratic programs, can obtain the optimal solution to these problem through optimality conditions. For other types of bilevel programming problems research has focused on designing swarm intelligence and hybrid algorithms, which are currently more effective algorithm frameworks and have achieved better solutions when tested against certain problems. Existing methods for solving the BLPP can be grouped into the following categories:

A). Classical approaches

Some classic methods for BLPP include the simplex [14], branch-and-bound method [15], descending gradient method [16], and penalty function methods [17–19]. Classical methods typically apply the optimality conditions of the follower to convert the BLPP into a single-level problem. Dempe [14]used the simplex method to propose an algorithm for solving linear BLPP. The algorithm introduces slack variables to find a basis as a feasible solution of the BLPP, through a simplex and iterative method. Although this method is more effective for solving small-scale linear BLPP it cannot be directly extended to nonlinear problems. Susanne et al. [16] replaced the the follower problem a with Karush-Kuhn-Tucker condition and applied the optimal value method to transform the original problem into a single-level problems. This approach could effectively satisfy the complementary conditions in the mathematical programming problem in Banach space and introduce M stationarity. Using the dual gap as a penalty function, the literature of White [17] proposed an effective algorithm for solving the BLPP. It adopts a newly designed precise penalty function method to obtain the global optimal solution of the linear situation and conducts a novel theoretical analysis. In the literature [18, 19], a weak linear BLPP dealt with the penalty function and Karush-Kuhn-Tucker conditions.

B). Evolutionary approaches

The evolutionary algorithms, as representatives of the swarm intelligence optimisation technology, have been widely used to solve various BLPP over the past few decades. An early evolutionary algorithm was proposed by Mathieu et al. [20], who applied linear programming methods to handle followers' problems and used a genetic algorithm to explore the search space of leaders' problems. Focused on BLPPs whose follower's problem is convex programming, Wang [21] proposed an evolutionary algorithm with an embedded constraint processing method. This method applies Karush-Kuhn-Tucker conditions to transform the followers' problems, turning the original model into a single-

level programming model. In addition, to obtain sufficient feasible solutions, a constraint-processing method was designed. Based on similar optimality conditions, Li [22] presented a genetic algorithm that solves nonlinear/linear fractional BLPP. In another study on the presence of the so-called pseudo-feasible solutions in evolutionary bilevel optimisation(QBCA-2), the focus was on determining how pseudo-feasible solutions can affect the performance of an evolutionary algorithm. Moreover, a novel and scalable set of test problems with characterised pseudo-feasible solutions was introduced in [23]. In the literature [24], a co-evolutionary algorithm was proposed to solve BLPP. In the evolution process, the follower problem was solved in two stages. Aboelnaga et al. [25] proposed an improved genetic algorithm and chaotic search method. Goshu et al. [26] proposed a metaheuristic algorithm for random BLPP. An evolutionary algorithm for solving nonlinear bilevel programming problems has been presented in the literature [27]. The algorithm is designed by reflecting the optimal solution of the follower problem to the leader problem. To ensure the quality of each iteration, the algorithm adaptive changes the population size during the evolution process and generates individuals using the tabu search method.

C). Hybrid approaches
The hybrid algorithm [28] is a common method used to solve BLPP. Abo-Elnaga et al. [29] proposed a multi-sine-cosine algorithm to solve nonlinear BLPP. The sine-cosine algorithms based on three different populations were presented. The first population was used to deal with the leader programming problem, while the second one addressed the follower programming problem. In addition, the Karush-Kuhn-Tucker condition was applied to transform the initial problem into a constrained optimisation problem, which was solved using the third population. If the objective function value is equal to zero, then the solution obtained by solving the leader and follower problems is deemed feasible. Wang [30] proposed a particle swarm distribution estimation algorithm with an embedded Estimation of Distribution Algorithm to solve nonlinear BLPP. Before executing the speed and location update rules in Partical Swarm Optimization, a Gaussian distribution was applied to generate offspring to replace some inferior individuals (particles) in the current population.

D). EA based on approximate methods
To avoid lengthy calculation processes, some evolutionary algorithms utilise approximation techniques to improve the efficiency of the intermediate calculation process. A bilayer covariance matrix adaptive evolution strategy(BLCMAES) is proposed in [31]. The method designed a sharing mechanism so that prior knowledge of the follower problem can be extracted from the leader optimizer, reduced the number of evaluations of the follower problem. Furthermore, an optimization-based elite retention mechanism is proposed to keep track of the elite and avoid incorrect solutions. Sinha et al. [32] proposed an evolutionary algorithm that uses approximate functions to adress BLPPs. For offspring generated by evolutionary operators, the approximation method can reduce the number of evaluations of the follower objective function. Using approximate Karush-Kuhn-Tucker conditions, Sinha et al. [33, 34] transformed the BLPP into a single-level optimisation problem; then leveraged an evolutionary algorithm embedded with the idea of neighbourhood measurements for the transformed model. Sinha et al. [35] presented an evolutionary optimisation algorithm (BLEAQ-2) that fits an extreme value mapping using the relationship between the leader and follower variables. Islam et al. [36] introduced an evolutionary algorithm that employs three types of surrogate models to approximate the optimal solution of the follower problem. Based on the linear programming

optimality conditions, Li [37] proposed a genetic algorithm for solving linear BLPP. In this method, the follower is adopted as the search object of the evolutionary algorithm, and the lower variables in the leader problems are replaced by possible solution functions. This process locally optimises the leader variables.

## Research motivation

In the field of engineering and economic management, optimization models involving different decision-making levels often appear, such problems are called multi-level optimization problems. Since the decision-making process of the problem is hierarchical, it is also called a hierarchical optimization problem. BLPP is a typical representative of multi-level hierarchical optimization problems, and has become an important research field for optimization problems due to its extensive practical application background and algorithmic challenges. Different from multi-objective optimization, the decision makers of BLPP are at two different levels, and this hierarchical structure often leads to problems that are non-convex and non-differentiable. Based on the above characteristics, it is often difficult for traditional optimization algorithms based on gradients to find the global optimal solution to the BLPP. Evolutionary algorithm be increasingly used to solve BLPP because it have the characteristics of global convergence, and there is no restriction on functions that are convex and differentiable.

In this paper, driven the correlation coefficient and surrogate model, an evolutionary algorithm (TCEA) is proposed to solve complex nonlinear BLPP. The algorithm has the following characteristics: First, the isodata clustering method [38] is used to group the initial populations, and then determine the correlation coefficients of the leader and follower objective functions in each group based on the rank of the leader (follower) objective function, after that, some points are selected and updated in the offsprings based on the correlation coefficient value in each group. Second, for the offspring individuals produced by the crossover and mutation operators, the surrogate model is used to approximate the solution of the follower programming problem, and so does to reduce the number of evaluations of follower problems.

The remainder of this paper is organized as follows. The basic concepts of BLPP are described in the next Section. The correlation coefficients and surrogate models are presented in Section 3. A new evolutionary algorithm based on the above methods are stated in Section 4. Experimental results and comparisons are provided in Section 5. We conclude our approach in Section 6.

## Basic concepts

Some basic definitions for problem (1) are summarized as follows:

1. Constraint region:

$$S = \{(x, y) | G(x, y) \leq 0, g(x, y) \leq 0, x \in R^n, y \in R^m\}$$

2. Follower's feasible set for $x$ fixed:

$$S(x) = \{y | y \in R^m, g(x, y) \leq 0\}$$

3. Projection of S onto the leader's decision space:

$$S(X) = \{x|\exists y, such\ that(x, y) \in S\}$$

4. Follower's rational reaction set for each $x \in S(x)$

$$M(x) = \{y|y \in argmin f(x, v), v \in S(x)\}$$

5. Inducible region:

$$IR = \{x, y) \in S|y \in M(x)\}$$

In terms of the aforementioned definitions, problem (1) can also be witten as

$$min\{F(x, y)|(x, y) \in IR\}$$

In order to ensure that problem (1) is well posed, in the remainder, we always assume that
(*A*1) S is nonempty and compact;
(*A*2) For all decisions taken by the leader's, the follower's has some room to react, that is, S
$(x) \neq \phi$;
(*A*3) The follower's problem has unique optimal solution for each fixed *x*.

## Main improvement schemes

### Correlation coefficients

The challenge faced in solving BLPP is that evaluating the follower programming problem involves a large amount of calculation. Therefore, to reduce the optimization times of the follower problem, we update a part of the population points by means of the relationship between the objective functions of the leader and follower (called correlation coefficients), thereby reducing the calculation times of the follower optimization problem. The correlation coefficients are defined as follows:

Given $z$ points, we acquire the leader and the follower objective function values $F(x_i, y_i)$ and $f(x_i, y_i)$, $i = 1, 2, \ldots, z$. Then, we sort the objective functions based on their values, and the sequence numbers are denoted $r_{F(x_i,y_i)}$ and $r_{f(x_i,y_i)}$, $i = 1, 2, \ldots, z$, respectively.

The sequence number difference between the sorted leader and follower objectives is obtained as follows:

$$d(x_i, y_i) = r_{F(x_i,y_i)} - r_{f(x_i,y_i)}, i = 1, 2, \ldots, z \tag{2}$$

Correlation coefficient is defined as follows:

$$\rho = 1 - \frac{6 \cdot \sum_{s=1}^{z} d(x_i, y_i)^2}{z \cdot (z^2 - 1)} \qquad \rho \in [-1, 1] \tag{3}$$

The larger the value of $\rho$ is, the more similar the changing trends of the objective functions of the leader and the follower are. Conversely, the changing trends of the objective functions of the leader and the follower are different. Particularly, if $\rho = 1$, the objective functions of the leader and follower exhibit exactly the same trends. When $\rho = -1$, the objective functions of the leader and the follower exhibit opposite trends. For example, we take $z = 5$ and set

$F(x_1, y_1) = 2.5, F(x_2, y_2) = 2.3, F(x_3, y_3) = 1.4, F(x_4, y_4) = 1.6, F(x_5, y_5) = 5.8$.
After sorting rank:
$F(x_3, y_3) = 1.4, F(x_4, y_4) = 1.6, F(x_2, y_2) = 2.3, F(x_1, y_1) = 2.5, F(x_5, y_5) = 5.8$.
$r_{F(x_1,y_1)} = 4, r_{F(x_2,y_2)} = 3, r_{F(x_3,y_3)} = 1, r_{F(x_4,y_4)} = 2, r_{F(x_5,y_5)} = 5$.
And given
$f(x_1, y_1) = 5.5, f(x_2, y_2) = 2.7, f(x_3, y_3) = 1.6, f(x_4, y_4) = 3.8, f(x_5, y_5) = 6.6$.
After sorting rank:
$f(x_2, y_2) = 1.6, f(x_3, y_3) = 2.7, f(x_1, y_1) = 3.8, f(x_5, y_5) = 5.5, f(x_4, y_4) = 6.6$.
$r_{f(x_1,y_1)} = 4, r_{f(x_2,y_2)} = 2, r_{f(x_3,y_3)} = 1, r_{f(x_4,y_4)} = 3, r_{f(x_5,y_5)} = 5$.
Then

$$d(x_1, y_1) = 0, d(x_2, y_2) = 1, d(x_3, y_3) = 0, d(x_4, y_4) = -1, d(x_5, y_5) = 0$$

and the correlation coefficients

$$\rho = 1 - \frac{6 \cdot (0^2 + 1^2 + 0^2 + (-1)^2 + 0^2)}{5 \cdot (5^2 - 1)} = 0.9$$

It turns to be that, the objective functions of the leader and the follower have the more similar the changing trends.

## Surrogate models

In BLPP, the procedure to finding a feasible solution results in a significant amount of computation in solving BLPP, particularly when the problem is large. And the optimal solutions to the follower's problem are always determined by the leader's variables. This means that the optimal solution of the follower problem is a function of the leader's variables. However, the function is often implicit and can not be obtained analytically. In the proposed approach, we take the polynomial fitting as surrogate models [39] to estimate the optimal solutions to the follower's problems.

The polynomial fitting demonstrates better performance in fitting unknown functions and can efficiently decrease the computational times of the follower problems. It is noteworthy that for these fitting points, each follower's variable value must be optimal when the leader's components are fixed. In the proposed algorithm, the polynomial fitting is generated as follows. First, an initial population of $N$ points $x_i$, $i = 1, 2, \ldots, N$ is gotten, the optimal solutions to the follower problem are denoted by $y_i$, $i = 1, 2, \ldots, N$. thus $N$ point pairs of can be obtained. These point pairs are used as fitting nodes to generate an polynomial curve.

$$y^j(x) = f^j(x), j = 1, 2, \ldots, m \tag{4}$$

where

$$y^j(x) = a_0 + a_1 x + a_2 x^2 + \ldots + a_k x^k \tag{5}$$

i.e. each of $y^j$, $j = 1, \ldots, m$, is a function of $x$ and $y(x) = (y^1, y^2, \ldots, y^m)$. Where $k$ is the highest degree of the polynomial, $a_0, a_1, a_2, \ldots, a_k$ is the undetermined coefficient and calculated by:

$$\begin{bmatrix} 1 & x_1 & \cdots & x_1^k \\ 1 & x_2 & \cdots & x_2^k \\ \vdots & \vdots & \vdots & \vdots \\ 1 & x_N & \cdots & x_N^k \end{bmatrix} \begin{bmatrix} a_0 \\ a_1 \\ \vdots \\ a_k \end{bmatrix} = \begin{bmatrix} y_1^j \\ y_2^j \\ \vdots \\ y_N^j \end{bmatrix} \tag{6}$$

According to the above-mentioned method, we can obtain the approximate optimal solutions to the follower problem.

## Proposed algorithm

In this manuscript, an evolutionary algorithm based on surrogate models and correlation coefficients, denoted by TCEA, is developed to solve BLPP. Fig 1 gives the flowchart of TCEA.

The detailed procedure of the proposed algorithm can be described as follows:

**Step 1** (Initial population)

The idea of uniform design [40] is adopted to produces $N$ points $x_i$, $i = 1, \ldots, N$, resulting in an initial population $pop(0)$ of size $N$. Set $gen = 0$, $D = \Phi$.

**Step 2** (Fitness evaluation)

For each $x_i$, solve the follower problems and obtain the optimal solution $y_i$, $i = 1, \ldots, N$. These points are put into $D$. The value of the leader objectives are taken as $F(x_i, y_i)$, $i = 1, \ldots, N$. Construct the polynomial fitting (surrogate models), as in Section 3.2. Use the isodata method to divide the generated points into $p$ groups, denoted as $I_1, I_2, \ldots, I_p$. Then take advantage of the correlation coefficients method in Section 3.1 to acquire the value of $\rho$ in each group, denoted by $\rho_1, \rho_2, \ldots, \rho_p$.

**Step 3** (Crossover)

For each crossover parent individual $x_i$, take an best individual as $\tilde{x}_i$, and perform the following crossover operator using the spherical search method:

Set: $r = max\{max_{1 \leq i \leq n}(\tilde{x}_i - l_i), max_{1 \leq i \leq n}(u_i - \tilde{x}_i) \times \alpha\}$ here $\alpha \in (0, 1)$ is a constant, which is called the shrinkage rate of the radius, and the method is as follows:

$$\begin{cases} \alpha = 10^{-10}, & max_{1 \leq i \leq n}(u_i - \tilde{x}_i) \leq 100 \\ \alpha = 10^{-8}, & 100 < max_{1 \leq i \leq n}(u_i - \tilde{x}_i) \leq 1000 \\ \alpha = 10^{-6}, & max_{1 \leq i \leq n}(u_i - \tilde{x}_i) > 1000 \end{cases} \tag{7}$$

$u_i$, $l_i$ is the leader and follower's bounds of $x_i$. Take a uniformly distributed value of $\theta_1, \theta_2, \ldots, \theta_J \in [0, 2\pi]$, $\beta_1, \beta_2, \ldots, \beta_J \in (-\pi/2, \pi/2)$ then use the spherical search method to generate a crossover operator as follows: (n is the dimension of the leader variable)

$$x_{2n-1}^j = \tilde{x}_{2n-1} + (\sqrt{2}r \setminus \sqrt{n})cos\theta_j \tag{8}$$

$$x_{2n}^j = \tilde{x}_{2n} + (\sqrt{2}r \setminus \sqrt{n})sin\theta_j \tag{9}$$

**Step 4** (Mutation)

Gaussian mutation is adopted. Suppose that $\bar{q}$ is an individual chosen for mutation, then the offspring $O\bar{q}$ of $\bar{q}$ is generated as follows:

$$O\bar{p} = \bar{q} + \Delta, \Delta \sim N(0, \sigma^2) \tag{10}$$

**Step 5** (Offspring population $pop'(gen)$)

For offspring set $(x_{o1}, x_{o2}, \ldots, x_{o\lambda})$ generated through the crossover and mutation operation. A surrogate model, the polynomial fitting, is used to obtain the approximated solutions $(y'_{o1}, y'_{o2}, \ldots, y'_{o\lambda})$ to the follower's program. We only update some of offspring points based on the value of $\rho_1, \rho_2, \ldots, \rho_p$ in each group as follows:

Case 1: If in the group $\tau$, $\tau = 1, 2, \ldots, p$, the value of $\rho_\tau$ is greater than the given threshold $\mu > 0$ (according to the results of the experiment), that is to say, this value is near to 1, it means that the leader's and the follower's objectives have the same changing trend. If the leader

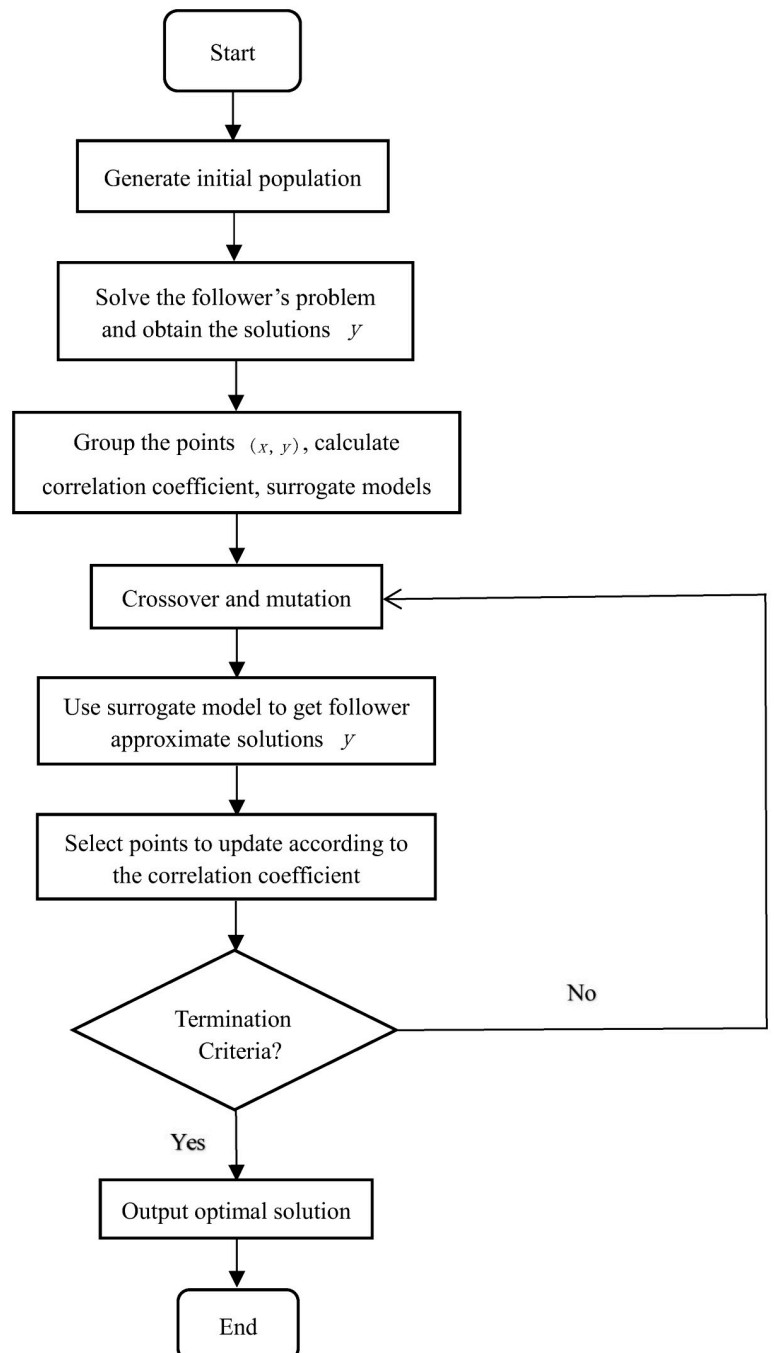

**Fig 1. The frame diagram of TCEA.**

objective function at point $(x_{oi}, y_{oi})$ satisfies $F(x_{oi}, y_{oi}) < \bar{F}$ (a predetermined threshold), it means that the leader's objective maybe become minimizer when the follower's objective is optimized. As a result, point $(x_{oi}, y_{oi})$ needs to be updated, that is to say, the follower problem is solved and the solution to the follower's problem is updated. If it is $F(x_{oi}, y_{oi}) > \bar{F}$ at point

$(x_{oi}, y_{oi})$, it means that the point is unpromising even if the follower's solution is updated. As a result, point $(x_{oi}, y_{oi})$ is not updated.

Case 2: If in the group $\tau$, $\tau = 1, 2, \ldots, p$, the value of $\rho_\tau$ is less than the given threshold $-\mu$ (according to the results of the experiment), i.e. The value is near to $-1$, it means that the leader and the follower objective functions have opposite trends. At this case, the leader's objective maybe become worse when the follower's objective is minimized. However, it is expected the worsen objective values are still better than the predetermined threshold $F^*$ (a bilevel feasible objective value). This means these points with small objective values are potential to be refined. The smallest objective is denoted by $F_{best}$ (maybe infeasible), and these points with objective value in $[F_{best}, F^*]$ should be further updated in a probabilistic sense. At point $(x_{oi}, y_{oi})$, if the objective satisfies that $F(x_{oi}, y_{oi}) < F^*$ and $F(x_{oi}, y_{oi}) \geq F_{best}$, then the point will be updated with probability $prob(oi)$.

$$prob(oi) = \frac{F^* - F(x_{oi}, y_{oi})}{F^* - F_{best}} \tag{11}$$

Obviously,

$$prob(oi) \in [0, 1]$$

Then, an offspring set $pop'(gen)$ with size $\eta$, $\eta \leq \lambda$, is obtained, and the values of the leader objective functions are $F(x_{oi}, y_{oi})$, $i = 1, 2, \ldots, \eta$. Put these $\eta$ solutions into archive set $D$.

**Step 6** (Selection)

Select the best $N$ individuals from set $pop(gen) \bigcup D$ to form the next generation of population $pop(gen + 1)$;

**Step 7** (Termination condition)

If the stopping criterion is satisfied, then stop and output the best one in set $D$; otherwise, set $gen = gen + 1$, go to Step 3.

## Simulation results

### Test examples

To demonstrate the feasibility and efficiency of the proposed algorithm TCEA, it was compared with the three already existing algorithms [23, 35, 31] developed for dealing with BLPP. Furthermore, we tested TCEA on six examples taken from the literature [23]; All six examples are presented as follows:

Example $F01$

$$\begin{cases} \min_x F(x, y) = | \sum_{i=1}^{k} \left( y_i - \frac{x_i^3}{100} \right)^2 - \sum_{i=1}^{k} x_i^2| + |10 + y_{k+1} + 10^6 \cdot \sum_{i=k+2}^{m} y_i^2 - \sum_{i=k+1}^{n} x_i^2| \\ \min_y f(x, y) = \sum_{i=1}^{k} \left( y_i - \frac{x_i^3}{100} \right)^2 - \sum_{i=1}^{k} x_i^2 + 10 + y_{k+1} + 10^6 \cdot \sum_{i=k+2}^{m} y_i^2 - \sum_{i=k+1}^{n} x_i^2 \end{cases}$$

Example *F*02

$$
\begin{cases}
\min_{x} F(x,y) = |(y_1 - x_1 sin(x_1))^2 + 10^6 \cdot \sum_{i=2}^{k}(y_i - sin(x_i))^2 - \sum_{i=1}^{k} x_i | \\
\qquad + |\sum_{i=k+1}^{m}(y_i - \sqrt{i})^2 + (\sum_{i=k+1}^{m} 0.5(y_i - \sqrt{i}))^2 \\
\qquad + (\sum_{i=k+1}^{m} 0.5(y_i - \sqrt{i}))^4 - x_{i+1}^2 - 10^6 \cdot \sum_{i=k+2}^{n} x_i^2 | \\
\min_{y} f(x,y) = (y_1 - x_1 sin(x_1))^2 + 10^6 \cdot \sum_{i=2}^{k}(y_i - sin(x_i))^2 - \sum_{i=1}^{k} x_i \\
\qquad + \sum_{i=k+1}^{m}(y_i - \sqrt{i})^2 + (\sum_{i=k+1}^{m} 0.5(y_i - \sqrt{i}))^2 \\
\qquad + (\sum_{i=k+1}^{m} 0.5(y_i - \sqrt{i}))^4 - x_{i+1}^2 - 10^6 \cdot \sum_{i=k+2}^{n} x_i^2
\end{cases}
$$

Example *F*03

$$
\begin{cases}
\min_{x} F(x,y) = |\sum_{i=1}^{k}\left(y_i - \frac{x_i^3}{100}\right)^2 - \sum_{i=1}^{k} x_i^2 | \\
\qquad + |\sum_{i=k+1}^{m-1}(100(y_i^2 - y_{i+1})^2 + (y_i - 1)^2) - \sum_{i=k+1}^{n} x_i^2 | \\
\min_{y} f(x,y) = \sum_{i=1}^{k}\left(y_i - \frac{x_i^3}{100}\right)^2 - \sum_{i=1}^{k} x_i^2 \\
\qquad + \sum_{i=k+1}^{m-1}(100(y_i^2 - y_{i+1})^2 + (y_i - 1)^2) - \sum_{i=k+1}^{n} x_i^2
\end{cases}
$$

Example *F*04

$$
\begin{cases}
\min_{x} F(x,y) = |\sum_{i=1}^{k}\left(y_i - \frac{10}{1 + 2.5x_i^2}\right)^2 - \sum_{i=1}^{k} x_i^2 | \\
\qquad + |10(m-k) + \sum_{i=k+1}^{m} y_i^2 - 10cos(2\pi y_i) - \sum_{i=k+1}^{n} x_i^2 | \\
\min_{y} f(x,y) = \sum_{i=1}^{k}\left(y_i - \frac{10}{1 + 2.5x_i^2}\right)^2 - \sum_{i=1}^{k} x_i^2 \\
\qquad + 10(m-k) + \sum_{i=k+1}^{m} y_i^2 - 10cos(2\pi y_i) - \sum_{i=k+1}^{n} x_i^2
\end{cases}
$$

Example *F*05

$$
\begin{cases}
\min_{x} F(x,y) = |10k + \sum_{i=1}^{k}((y_i - x_i)^2 - 10cos(2\pi|y_i - x_i|)) - \sum_{i=1}^{k} x_i^2 | \\
\qquad + |\sum_{i=k+1}^{m}|y_i|^{i-k+1} - \sum_{i=k+1}^{n} x_i^2 | \\
\min_{y} f(x,y) = 10k + \sum_{i=1}^{k}((y_i - x_i)^2 - 10cos(2\pi|y_i - x_i|)) - \sum_{i=1}^{k} x_i^2 \\
\qquad + \sum_{i=k+1}^{m}|y_i|^{i-k+1} - \sum_{i=k+1}^{n} x_i^2
\end{cases}
$$

Example *F*06

$$
\begin{cases}
\min_{x} F(x,y) = |1 + \frac{1}{4}\sum_{i=1}^{k}(y_i - x_i)^2 - \prod_{i=1}^{k} cos\left(\frac{10(y_i - x_i)}{\sqrt{i}}\right) - \sum_{i=1}^{k} x_i^2 | + |10(m-k) \\
\qquad + \sum_{i=k+1}^{m} y_i^2 - 10cos(2\pi y_i) - 10(n-k) + \sum_{i=k+1}^{n}(x_i^2 - 10cos(2\pi x_i)) | \\
\min_{y} f(x,y) = 1 + \frac{1}{4}\sum_{i=1}^{k}(y_i - x_i)^2 - \prod_{i=1}^{k} cos\left(\frac{10(y_i - x_i)}{\sqrt{i}}\right) - \sum_{i=1}^{k} x_i^2 + 10(m-k) \\
\qquad + \sum_{i=k+1}^{m} y_i^2 - 10cos(2\pi y_i) - 10(n-k) + \sum_{i=k+1}^{n}(x_i^2 - 10cos(2\pi x_i))
\end{cases}
$$

## Parameter settings

In order to compare with the experimental data in the literature, the selected parameters are consistent with those in [23]. When the leader's and follower's variables are 5-dimensional, the parameters are chosen as follows:

Leader's variable dimension:$n = 2$; Follower's variable dimension:$m = 3$;

Popsize:$N = 25$; Maximum running algebra:$G_{max} = 50$;

Mutation probability:$Pm = 0.1$; Crossover probability:$Pc = 0.8$;

Number of runs:$Numrun = 10$; $k = 1$;

When the leader's and follower's variables are 10-dimensional, the parameters are chosen as follows:

Leader's variable dimension:$n = 5$; Follower's variable dimension:$m = 5$;

Popsize:$N = 50$; Maximum running algebra:$G_{max} = 50$;

Mutation probability:$Pm = 0.1$; Crossover probability:$Pc = 0.8$;

Number of runs:$Numrun = 10$; $k = 2$;

When the leader's and follower's variables are 20-dimensional, the parameters are chosen as follows:

Leader's variable dimension:$n = 10$; Follower's variable dimension:$m = 10$;

Popsize:$N = 100$; Maximum running algebra:$G_{max} = 100$;

Mutation probability:$Pm = 0.1$; Crossover probability:$Pc = 0.8$;

Number of runs:$Numrun = 10$; $k = 5$;

The optimal solution obtained by the algorithm TCEA is recorded as: $(x^*, y^*)$, the leader's and follower's objective function values are respectively denoted as:$F(x^*, y^*)$ and $f(x^*, y^*)$.

## Result analysis

We executed an algorithm on a computer (Intel(R) Core(TM)i5-8250U CPU@ 160 GHz 1.80 GHz)using the MATLAB. For all six examples, TCEA was compared with the three algorithms in the literature [23, 31, 35]. Tables 1–3 shows the average value of optimal results running 10 times independently for all examples, and we calculated the results of dimension 5, dimension 10 and dimension 20. In order to facilitate the comparison of TCEA, we take the same stopping criterion as that in the literature, that is: the evaluation times of the objective function of the leader's is the stopping criterion of TCEA. Among them, when the variable dimensions are 5 and 10, the evaluation times of the leader's objective function are 2500 and 3500, respectively. The optimal solutions obtained are listed in Tables 1–3.

**Table 1. Comparison between the results obtained by TCEA and the real objective values in the case of 5 dimensions.**

| Test problem | $Real - (F(x^*, y^*), f(x^*, y^*))$ | $((F(x^*, y^*), f(x^*, y^*))$ | $(x^*, y^*)$ |
|:---:|:---:|:---:|:---:|
| F01 | (0.0000, 0.0000) | (0.0000, 0.0000) | $x^* = (0.0000, 0.0000)$<br>$y^* = (0.0000, -10.0000, 0.0000)$ |
| F02 | (0.0000, 0.0000) | (0.0000, 0.0000) | $x^* = ((0.0000, 0.0000)$<br>$y^* = (0.0000, 1.4142, 1.7318)$ |
| F03 | (0.0000, 0.0000) | (0.0000, 0.0000) | $x^* = -0.0033, -0.0049)$<br>$y^* = (0.0000, 1.0000, 1.0000)$ |
| F04 | (0.0000, 0.0000) | (0.0000, 0.0000) | $x^* = (0.0000, 0.0000)$<br>$y^* = (10.0000, 10.0000, 0.0000)$ |
| F05 | (0.0000, 0.0000) | (0.0000, 0.0000) | $x^* = (0.0000, 0.0000)$<br>$y^* = (0.0000, 0.0000, 0.0000)$ |
| F06 | (0.0000, 0.0000) | (0.0000, 0.0000) | $x^* = (0.0008, 0.0000)$<br>$y^* = (0.0007, 0.0000, 0.0000)$ |

**Table 2. Comparison between the results obtained by TCEA and the real objective values in the case of 10 dimensions.**

| Test problem | $Real - (F(x^*, y^*), f(x^*, y^*))$ | $((F(x^*, y^*), f(x^*, y^*))$ | $(x^*, y^*)$ |
|---|---|---|---|
| F01 | (0.0000, 0.0000) | (0.0043, −0.0043) | $x^* = (0.0367, -0.0312, 0.0293, 0.0325, 0.0073)$<br>$y^* = (0.0000, 0.0000, -10.0000, 0.0000, 0.0000)$ |
| F02 | (0.0000, 0.0000) | (0.0000, 0.0000) | $x^* = ((0.0001, 0.0000, -0.0001, 0.0000, 0.0000)$<br>$y^* = (0.0000, 0.0000, 1.7318, 1.9998, 2.2361)$ |
| F03 | (0.0000, 0.0000) | (0.0098, −0.0098) | $x^* = (-0.0282, 0.0217, -0.0375, -0.0706, 0.0467)$<br>$y^* = (0.0000, 0.0000, 1.0000, 1.0000, 1.0000)$ |
| F04 | (0.0000, 0.0000) | (0.0000, 0.0000) | $x^* = (-0.0019, 0.0044, 0.0016, 0.0039, 0.0027)$<br>$y^* = (10.0000, 10.0000, 0.0000, 0.0000, 0.0000)$ |
| F05 | (0.0000, 0.0000) | (0.0000, −0.0000) | $x^* = (-0.0030, 0.0017, -0.0050, -0.0023, 0.0035)$<br>$y^* = (0.0030, 0.0017, -0.0000, 0.0000, 0.0000)$ |
| F06 | (0.0000, 0.0000) | (0.0056, 0.0000) | $x^* = (0.0169, 0.0160, -0.0509, 0.0100, 0.0096)$<br>$y^* = (0.0100, 0.0100, 0.0000, 0.0000, 0.0000)$ |

When the problem's dimension is 5, we can draw it from Table 1 that TCEA can find the same optimal value as the analytical solution on all test cases; When the dimension of the variable is 10, as can be seen from Table 2, for case F02, F04, and F05, TCEA can get the same optimal value as the analytical solution. In case F01 and F06, TCEA can attain the optimal value with a small error comparing with the analytical solution; In the case of variable dimensions are 20, in order to shorten the search time, we shrink the corresponding search space to half of the original search space, which is aimed at testing the computational savings of the algorithm on medium-scale problems in a short period of time. From Table 3, as far as the leader's

**Table 3. Comparison between the results obtained by TCEA and the real objective values in the case of 20 dimensions.**

| Test problem | $Real - (F(x^*, y^*), f(x^*, y^*))$ | $((F(x^*, y^*), f(x^*, y^*))$ | $(x^*, y^*)$ |
|---|---|---|---|
| F01 | (0.0000, 0.0000) | (0.0515, −0.0515) | $x^* = (-0.0503, 0.0754, -0.0500, 0.0420, -0.0187, -0.1244,$<br>$-0.1043, -0.0767, -0.0058, 0.0803)$<br>$y^* = (-0.0001, 0.0002, -0.0001, 0.0000, 0.0000, -10.0000,$<br>$0.0000, 0.0000, 0.0000, 0.0000)$ |
| F02 | (0.0000, 0.0000) | (0.2393, 0.2393) | $x^* = (0.0008, 0.0007, -0.0002, 0.0007, -0.0012, 0.0009,$<br>$-0.0003, -0.0002, 0.0002, -0.0002)$<br>$y^* = (0.0000, 0.0000, 0.0000, 0.0000, 0.0000, 2.4515, 2.6435,$<br>$2.8248, 3.0006, 3.1622)$ |
| F03 | (0.0000, 0.0000) | (0.0118, −0.0118) | $x^* = (-0.0127, -0.0330, 0.0025, -0.0812, 0.0091, 0.0516,$<br>$0.0214, 0.0126, -0.0081, -0.0234)$<br>$y^* = (0.0000, 0.0000, 0.0000, 0.0000, 0.0000, 1.0000, 1.0000,$<br>$1.0000, 1.0000, 1.0000)$ |
| F04 | (0.0000, 0.0000) | (0.0001, −0.0001) | $x^* = (-0.0034, -0.0018, -0.0060, 0.0051, -0.0005, 0.0018,$<br>$-0.0008, -0.0068, 0.0004, -0.0034)$<br>$y^* = (9.9997, 9.9999, 9.9991, 9.9994, 10.0000, -0.0000,$<br>$-0.0000, -0.0000, -0.0000, -0.0000)$ |
| F05 | (0.0000, 0.0000) | (0.0001, 38.7202) | $x^* = (-0.0007, 0.0067, 0.0038, -0.0034, 0.0027, 0.0033,$<br>$-0.0042, 0.0045, -0.0003, -0.0007)$<br>$y^* = (-0.0007, 0.0067, 0.0038, -0.0034, 0.0027, 0.0000, 0.0000,$<br>$0.0000, 0.0000, 0.0000)$ |
| F06 | (0.0000, 0.0000) | (0.7544, 0.7455) | $x^* = (-0.0029, -0.0088, -0.0082, -0.0082, -0.0067, 0.0005,$<br>$0.0011, 0.0035, 0.0023, -0.0020)$<br>$y^* = (-0.0029, -0.0086, -0.0081, -0.0082, -0.0068, 0.0000,$<br>$0.0000, 0.0000, 0.0000, 0.0000)$ |

**Table 4. Success rates of QBCA-2, BLCMAES, BLEAQ-2 and TCEA on 5, 10 and 20-dimensional test problems.**

| Dimension | Instance | QBCA − 2 | BLCMAES | BLEAQ − 2 | TCEA | +/−/≈ |
|---|---|---|---|---|---|---|
| 5 − dimension | F01 | 100% | 100% | 3% | 100% | 1/0/2 |
| | F02 | 100% | 100% | 100% | 100% | 0/0/3 |
| | F03 | 100% | 100% | 81% | 100% | 1/0/2 |
| | F04 | 97% | 84% | 81% | **100**% | 3/0/0 |
| | F05 | 97% | 100% | 91% | 100% | 2/0/1 |
| | F06 | 94% | 10% | 20% | **100**% | 3/0/0 |
| 10 − dimension | F01 | 100% | 87% | 3% | 100% | 2/0/1 |
| | F02 | 100% | 0% | 0% | 100% | 2/0/1 |
| | F03 | 100% | 100% | 0% | 100% | 1/0/2 |
| | F04 | 100% | 36% | 0% | 100% | 2/0/1 |
| | F05 | 100% | 68% | 3% | 100% | 2/0/1 |
| | F06 | 0% | 0% | 0% | **100**% | 3/0/0 |
| 20 − dimension | F01 | – | – | – | 60% | – |
| | F02 | – | – | – | 65% | – |
| | F03 | – | – | – | 60% | – |
| | F04 | – | – | – | 90% | – |
| | F05 | – | – | – | 90% | – |
| | F06 | – | – | – | 30% | – |

objective value is concerned, in Examples F01, F03, F04, and F05, the optimal solution of the leader objective function with small error corresponding to the analytical solution can be found. However, for the calculation examples F02 and F06, the error of the leader objective function value corresponding to the analytical solution is relatively large, which means that TCEA requires more calculation algebra.

In addition, in order to illustrate the performance of the proposed algorithm, the success rate of runs and an index of performance measurement, is introduced in algorithms. If the difference between the leader's objective value obtained by TCEA and the known analytic solution in one run is less than $\varepsilon_1(\varepsilon_1 = 1 \times 10^{-2})$, then the algorithm is successful. The value of $\varepsilon_1$ is consistent with the literature [23], and the success rate is defined as the number of successful runs dominates the total number of independent runs.

Table 4 shows the results of 31 times independent runs. We recorded the success rate of finding the optimal solutions of the leader objective functions. It can be seen from Table 4 that when the dimension is 5, the results of Examples F04 and F06 are better than the other three algorithms, and when the dimension is 10, the success rate of example F06 are also superior to the algorithms in the literature. In addition, TCEA has also been used to test the case when the variable is 20-dimensional. TCEA is successful in the tested case, except cases F02 and F06.

The symbols "+", "−" and ≈ indicate that the computational result is better than, worse than, and almost equal to that obtained by our algorithm, respectively. The best results are highlighted in bold in Table 4.

Tables 5–7 show the median and standard deviation(Std) in 10 runs by the TCEA when the variable dimensions are 5, 10, and 20. Meanwhile, the computational results are compared with those in the literature [23] for 5-dimensional and 10-dimensional cases. UL and LL accuracy statistics stand for the objective values obtained at the leader's and the follower's levels, respectively.

To facilitate comparison and illustrate the effectiveness of the TCEA, when the variable is 5 and 10 dimensions, we multiplied both the median and standard deviation (Std) by $1 \times 10^5$ in

**Table 5. UL and LL accuracy statistics from 31 independent runs by QBCA2, BLCMAES, BLEAQ-2 and TCEA on 5-dimensional test problems.**

| Median | QBCA2 | BLCMAES | BLEAQ-2 | TCEA | +/−/≈ |
|---|---|---|---|---|---|
| F01-UL | 4.31E-04 | 4.87E-07 | 3.64E-03 | 3.6089E-06 | 2/1/0 |
| F02-UL | 3.83E-04 | 6.14E-07 | 3.42E-07 | 3.8025E-07 | 1/0/2 |
| F03-UL | 4.94E-04 | 5.29E-07 | 5.98E-04 | 4.4321E-07 | 2/0/1 |
| F04-UL | 5.24E-04 | 4.83E-07 | 9.50E-04 | 6.2367E-05 | 2/1/0 |
| F05-UL | 5.57E-04 | 5.36E-07 | 6.89E-04 | 1.0013E-07 | 2/0/1 |
| F06-UL | 4.08E-04 | 9.95E-01 | 9.71E-03 | 4.9845E-03 | 1/1/1 |
| F01-LL | 4.31E-04 | 4.87E-07 | 3.51E-03 | 3.6089E-06 | 2/1/0 |
| F02-LL | 3.83E-04 | 6.14E-07 | 3.41E-07 | 3.8025E-07 | 1/0/2 |
| F03-LL | 4.94E-04 | 5.29E-07 | 4.41E-04 | 4.4321E-07 | 2/0/1 |
| F04-LL | 5.24E-04 | 4.83E-07 | 4.97E-04 | 6.2367E-05 | 2/1/0 |
| F05-LL | 5.57E-04 | 5.36E-07 | 2.86E-04 | 1.5000E-07 | 2/0/1 |
| F06-LL | 4.08E-04 | 9.95E-01 | 3.81E-03 | 4.9845E-03 | 1/1/1 |
| Std | QBCA2 | BLCMAES | BLEAQ-2 | TCEA | +/−/≈ |
| F01-UL | 2.52E-04 | 2.78E-07 | 3.15E-02 | 2.6884E-05 | 2/1/0 |
| F02-UL | 2.92E-04 | 2.75E-07 | 2.42E-04 | 2.5533E-05 | 2/1/0 |
| F03-UL | 2.89E-04 | 2.59E-07 | 2.23E-03 | 2.3438E-07 | 2/0/1 |
| F04-UL | 2.34E-03 | 3.21E-07 | 3.89E-02 | 1.2666E-05 | 2/1/0 |
| F05-UL | 2.55E-04 | 2.70E-07 | 1.77E-03 | 1.0013E-07 | 2/0/1 |
| F06-UL | 1.38E-03 | 4.85E-01 | 2.11E-01 | **6.5971E-04** | 3/0/0 |
| F01-LL | 2.52E-04 | 2.78E-07 | 3.16E-02 | 2.6884E-05 | 2/1/0 |
| F02-LL | 2.92E-04 | 2.75E-07 | 2.42E-04 | 2.5533E-05 | 2/1/0 |
| F03-LL | 2.89E-04 | 2.59E-07 | 2.12E-03 | 2.3438E-07 | 2/0/1 |
| F04-LL | 2.34E-03 | 3.21E-07 | 3.91E-02 | 1.2666E-05 | 2/1/0 |
| F05-LL | 2.55E-04 | 2.70E-07 | 3.88E-04 | 1.5000E-07 | 2/0/1 |
| F06-LL | 2.91E-01 | 4.85E-01 | 1.90E-01 | **6.5971E-04** | 3/0/0 |

Tables 5 and 6, and stipulated that the generated values greater than or equal to 100 are assigned 100. Figs 2 and 3 show the histogram corresponding to the median and Std in Table 5 for the case of 5-dimensional variables, and Figs 4 and 5 display the histogram corresponding to Median and Std in Table 6 for the case of 10-dimensional variable, respectively.

It can be seen from Fig 2 that under the case of 5-dimensional variables, the median corresponding to the leader's and the follower's objective values are more effective than other methods in F01-03, and F05. Meanwhile, as can be seen from Fig 3, the Std corresponding to the leader and the follower objective function values of our algorithm in F03 and F05 is equivalent to the algorithm BLCMAES, but it is superior to the other two algorithms. On Problem F06, our algorithm is better than the other methods, and the advantage effect is obvious on Problems F03 and F05.

As seen in Fig 4, for the problems with 10-dimensional variable, the medians corresponding to the leader's and the follower's objectives are more effective than other methods in Problems F02, F03, and F06. As shown in Fig 5, the Std corresponding to the leader and the follower objective functions obtained by TCEA are better than those by other compared algorithms on F01, F02, F04 and F06.

It is quite difficult to solve the BLPP because we have to consider the hierarchy of the problem. Therefore, the amount of calculation required to solve the BLPP is very large. In the proposed algorithm, both an approximate computation scheme and correlation coefficients are adopted to reduce the computational cost caused by the follower's optimization. To illustrate

**Table 6. UL and LL accuracy statistics from 31 independent runs by QBCA2, BLCMAES, BLEAQ-2 and TCEA on 10-dimensional test problems.**

| Median | QBCA2 | BLCMAES | BLEAQ-2 | TCEA | +/−/≈ |
|--------|-------|---------|---------|------|-------|
| F01-UL | 7.48E-04 | 1.33E-05 | 1.51E-02 | 1.2558E-05 | 2/0/1 |
| F02-UL | 7.65E-04 | 2.21E+01 | 2.62E+01 | **4.2231E-05** | 3/0/0 |
| F03-UL | 7.58E-04 | 6.94E-07 | 1.63E-02 | 5.0401E-07 | 2/0/1 |
| F04-UL | 7.23E-04 | 1.48E-05 | 1.80E-01 | 1.5456E-04 | 1/1/1 |
| F05-UL | 7.97E-04 | 8.38E-07 | 5.65E-02 | 1.8483E-04 | 1/1/1 |
| F06-UL | 5.91E-01 | 9.95E-01 | 3.56E+00 | **1.5953E-03** | 3/0/0 |
| F01-LL | 7.48E-04 | 1.33E-05 | 1.18E-02 | 1.2558E-05 | 2/0/1 |
| F02-LL | 7.65E-04 | 2.21E+01 | 1.57E+01 | **4.2231E-05** | 3/0/0 |
| F03-LL | 7.58E-04 | 6.94E-07 | 1.63E-02 | 5.0401E-07 | 2/0/1 |
| F04-LL | 7.23E-04 | 1.48E-05 | 1.74E-01 | 1.5456E-04 | 1/1/1 |
| F05-LL | 7.97E-04 | 8.38E-07 | 2.71E-02 | 1.8483E-04 | 1/1/1 |
| F06-LL | 3.77E-01 | 9.95E-01 | 3.56E+00 | **1.5953E-03** | 3/0/0 |
| Std | QBCA2 | BLCMAES | BLEAQ-2 | TCEA | +/−/≈ |
| F01-UL | 2.17E-04 | 4.09E+00 | 1.22E-01 | **1.1127E-05** | 3/0/0 |
| F02-UL | 2.30E-04 | 3.09E+02 | 2.42E+01 | **3.3947E-05** | 3/0/0 |
| F03-UL | 2.12E-04 | 1.94E-07 | 8.41E-02 | 6.6445E-05 | 2/1/0 |
| F04-UL | 2.70E-04 | 7.69E-02 | 5.51E-01 | **9.6662E-05** | 3/0/0 |
| F05-UL | 1.86E-04 | 1.34E-05 | 1.08E-01 | **1.9441E-04** | 1/1/1 |
| F06-UL | 7.97E-01 | 1.09E+00 | 3.52E+00 | **2.0507E-02** | 3/0/0 |
| F01-LL | 2.17E-04 | 4.09E+00 | 1.20E-01 | **1.1127E-05** | 3/0/0 |
| F02-LL | 2.30E-04 | 3.09E+02 | 2.44E+01 | **3.3947E-05** | 3/0/0 |
| F03-LL | 2.12E-04 | 1.94E-07 | 8.06E-02 | 6.6445E-05 | 2/1/0 |
| F04-LL | 2.70E-04 | 7.69E-02 | 3.81E-01 | **9.6662E-05** | 3/0/0 |
| F05-LL | 1.86E-04 | 1.34E-05 | 9.78E-02 | 1.9441E-04 | 1/1/1 |
| F06-LL | 8.29E-01 | 1.09E+00 | 3.52E+00 | **2.0507E-02** | 3/0/0 |

the efficiency of the proposed algorithm on the above computational examples, we execute the proposed algorithm using two kinds of the follower solution methods. One updates all off-spring, whereas the other is to update only a part of them. The two algorithms stop once the optimal solutions are found. We recorded the CPU time for the purpose of comparison in Table 8 for cases of 5, 10 and 20-dimensional variables.

OMCPU represents CPU time needed by the method of updating all offspring, whereas CPU stands for the computational time cost by the method of only updating a part of individuals. Table 8 reveals that the proposed algorithm can save computational cost effectively for each example, which indicates that the proposed approximate scheme is efficient.

**Table 7. UL and LL accuracy statistics from 31 independent runs by TCEA on 20-dimensional test problems.**

| Test problems | Median-UL | Median-LL | Std-UL | Std-LL |
|---------------|-----------|-----------|--------|--------|
| F01 | 3.2173E-01 | 3.2173E-01 | 1.1127E-01 | 1.1127E-01 |
| F02 | 1.5175E-01 | 1.5175E-01 | 3.3947E-01 | 3.3947E-01 |
| F03 | 3.1980E-02 | 3.1980E-02 | 6.6445E-01 | 6.6445E-01 |
| F04 | 3.4658E-01 | 3.4658E-01 | 9.6662E-02 | 9.6662E-02 |
| F05 | 1.9597E-03 | 1.9597E-03 | 8.9441E-02 | 8.9441E-02 |
| F06 | 7.830E-01 | 7.830E-01 | 2.0507E-02 | 2.0507E-02 |

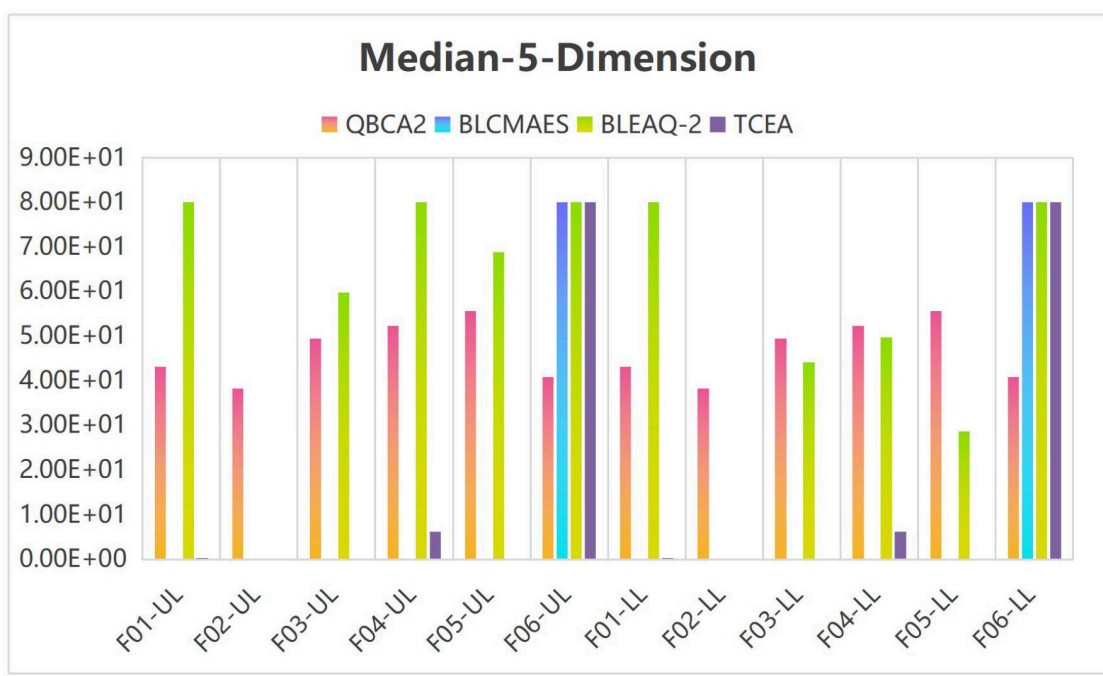

**Fig 2. Histogram of the median values on 5-dimensional problems.**

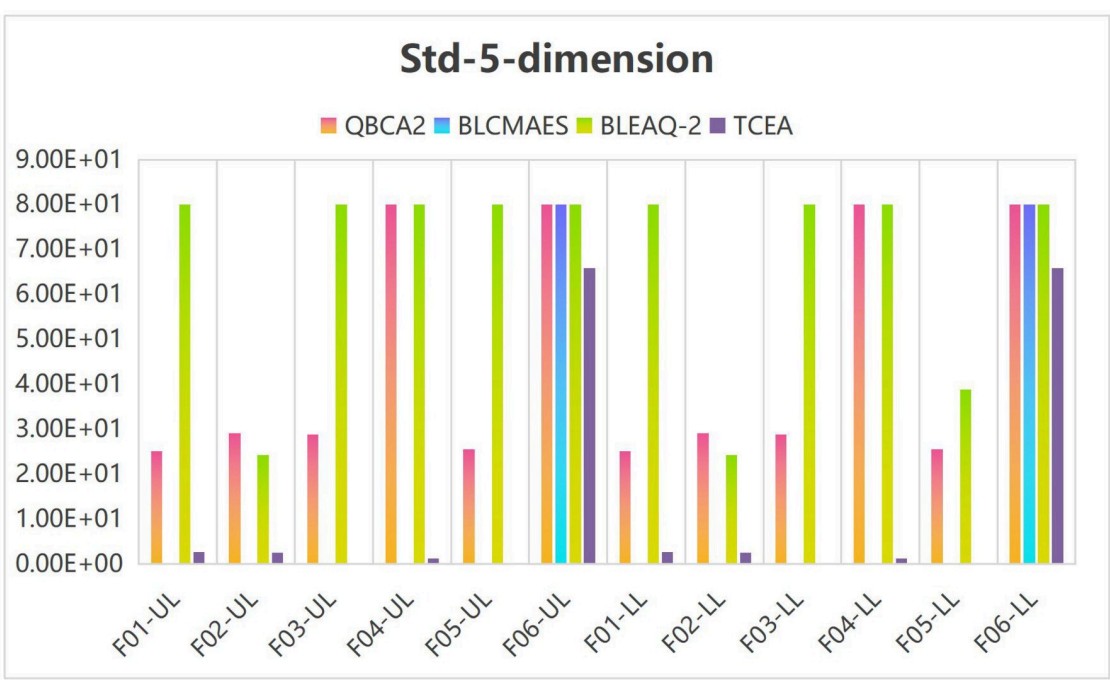

**Fig 3. Histogram of the Std values on 5-dimensional problems.**

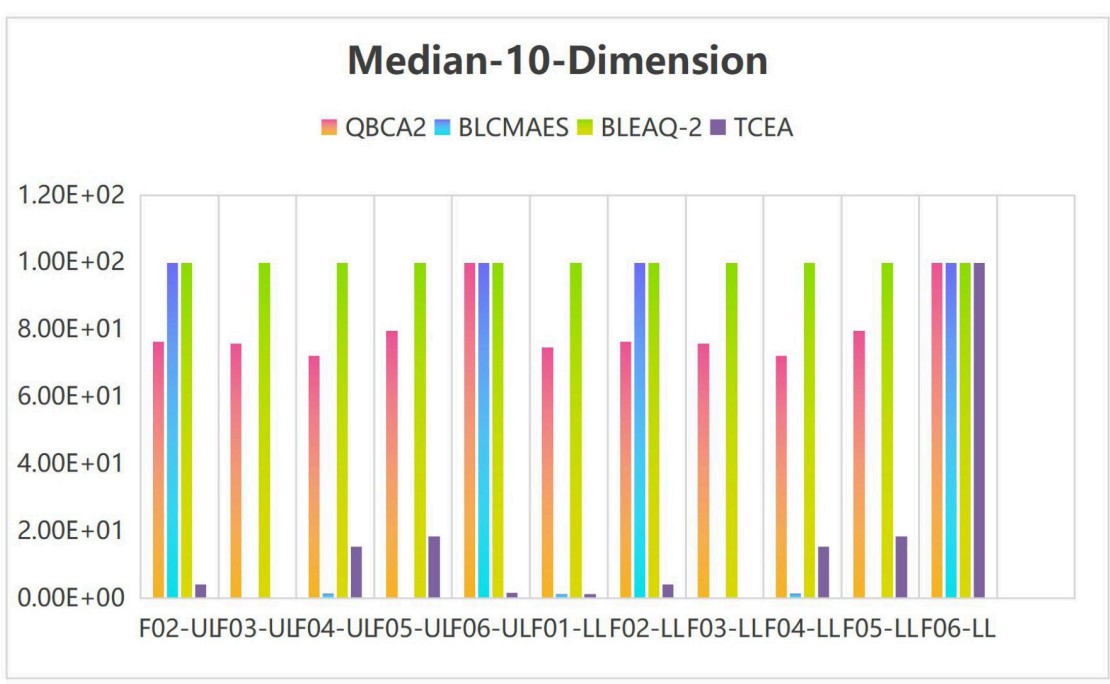

**Fig 4. Histogram of the median values on 10-dimensional problems.**

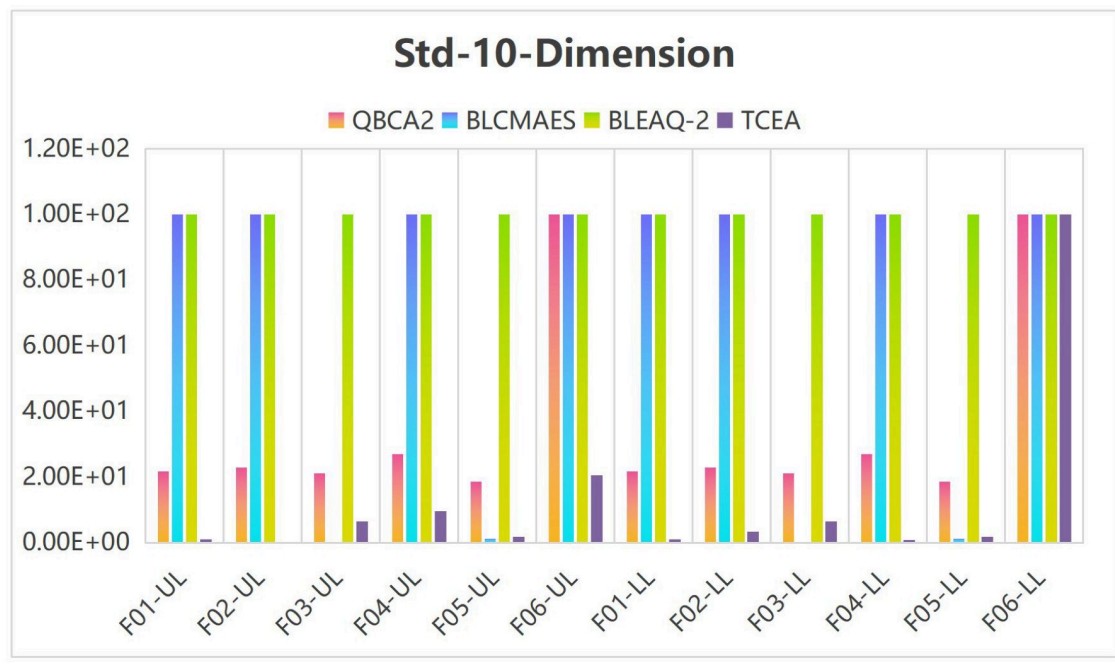

**Fig 5. Histogram of the Std values on 10-dimensional problems.**

**Table 8. Comparison of CUP time on 5, 10 and 20-dimensional test problems.**

| Test problems | OMCPU(s) 5-dimension | CPU(s) 5-dimension | OMCPU(s) 10-dimension | CPU(s) 10-dimension | OMCPU (s) 20-dimension | CPU(s) 20-dimension |
|---|---|---|---|---|---|---|
| F01 | 76.3918 | **25.6869** | 320.2124 | **147.2344** | 1468.8002 | **511.7339** |
| F02 | 68.1150 | **19.4717** | 316.9745 | **146.1719** | 1324.3715 | **421.5966** |
| F03 | 60.6603 | **19.5352** | 357.3386 | **153.7178** | 1765.9101 | **456.8823** |
| F04 | 79.0735 | **25.6744** | 330.0804 | **144.2656** | 1934.5218 | **508.3447** |
| F05 | 86.4893 | **23.6563** | 365.4467 | **150.8125** | 1518.6654 | **413.2659** |
| F06 | 77.4086 | **17.4531** | 337.4687 | **156.4531** | 2009.7399 | **514.8885** |

## Conclusion

BLPP is one of the hardest optimization models because it always accumulates the computational complexity of the hierarchical structure. Solving nonlinear BLPP is more challenging than solving linear BLPP. In this paper, we study a class of nonlinear BLPP problems, therefore, it has certain difficulties in both the accuracy of the solutions and the amount of calculations. Three efficient techniques are embedded in the proposed algorithm to improve the accuracy of the solutions and reduce the computational cost of the problem. One is the correlation coefficient method used to select the offspring obtained through crossover and mutation, this technique can appropriately reduce the computational complexity of solving the follower problem, that is to say it is can save a lots of CPU time of the algorithm. The other is the surrogate model which can efficiently reduce the computational cost of obtaining bilevel feasible solutions. The last technique is that the designed crossover and mutation operators can make the offspring individuals develop in a better direction, thereby improving the accuracy of the solutions. The simulation results in six computational examples show the efficiency of the proposed algorithm.

In the future research, the proposed algorithm will be adjusted and applied to solve some real-world problems which can be modeled as BLPP.

## Acknowledgments

We thank the editors and the anonymous reviewers for their professional and valuable suggestions.

## Author Contributions

**Data curation:** Yuhui Liu, Huafei Chen.

**Investigation:** Mei Ma.

**Methodology:** Yuhui Liu, Hecheng Li.

**Writing – original draft:** Yuhui Liu.

**Writing – review & editing:** Yuhui Liu.

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
