## [Decision Letter · Decision Letter 0]

23 Jun 2022

PONE-D-22-04037An evolutionary algorithm based on an approximation method and related techniques to solve bilevel programming problemsPLOS ONE

Dear Dr. Liu,

Thank you for submitting your manuscript to PLOS ONE. After careful consideration, we feel that it has merit but does not fully meet PLOS ONE’s publication criteria as it currently stands. Therefore, we invite you to submit a revised version of the manuscript that addresses the points raised during the review process.

We look forward to receiving your revised manuscript.

Kind regards,

Seyedali Mirjalili

Academic Editor

PLOS ONE

Journal Requirements:

4. Please ensure that you refer to Figure 1 in your text as, if accepted, production will need this reference to link the reader to the figure.

Reviewers' comments:

Reviewer's Responses to Questions

**Comments to the Author**

1. Is the manuscript technically sound, and do the data support the conclusions?

Reviewer #1: Yes

Reviewer #2: Yes

2. Has the statistical analysis been performed appropriately and rigorously? 

Reviewer #1: Yes

Reviewer #2: Yes

3. Have the authors made all data underlying the findings in their manuscript fully available?

Reviewer #1: No

Reviewer #2: Yes

4. Is the manuscript presented in an intelligible fashion and written in standard English?

Reviewer #1: Yes

Reviewer #2: Yes

5. Review Comments to the Author

Reviewer #1: The paper presents an interesting approach to improve the evolutionary algorithm for solving the complex nonlinear optimization problems. The paper is in the scope of PLOSE ONE and it has some strengths which can be it publishable. However, it has some weakness that should be overcome:

• Section Basic concepts is not clear enough. Please re-write this section.

• The application of the new proposed model needs to be supplemented.

• There are too many abbreviations in the full text, which is not conducive to readers' reading. It is recommended to optimize.

• Some abbreviations in the first time don't explain the meaning.

• Fig 1 is not good, please change it. I recommend the authors to have a look at the following works:

“Efficient method using Whale Optimization Algorithm for reliability-based design optimization of labyrinth spillway” or “A hybrid teaching–learning slime mould algorithm for global optimization and reliability-based design optimization problems “or even “Accurate Structural Reliability Analysis Using an Improved Line-Sampling-Method-Based Slime Mold Algorithm”.

• Some equations haven’t number. Line 268, 283, etc. It is recommended to identify the equations through numbering.

• The quality of Figs. 2-5 needs to improve.

• Conclusions should be more forceful and elaborated.

Good Luck

Reviewer #2: The paper proposes An evolutionary algorithm based on an approximation method and related techniques to solve bilevel programming problems. This is an interesting study, and the paper is generally well written and structured.

However, in my opinion, the paper has some shortcomings:

English needs revision by a native speaker.

The pros and cons of related work should be summarized in the introduction section, in order to highlight your work's novelty.

The real problem is not clearly identified. Authors are required to critically analyze the limitations and difficulties in previous studies, add some discussion on the existing methods and highlight what are their strengths and weaknesses so that justifies the development of a new model.

The materials for the literature review are not very well arranged to justify the proposed solution.

Results and discussions do not clearly support the claimed contribution.

Conclusion has to be improved.

6. PLOS authors have the option to publish the peer review history of their article (what does this mean?). If published, this will include your full peer review and any attached files.

Reviewer #1: No

Reviewer #2: **Yes: **nima khodadadi

---

## [Author Response · Author response to Decision Letter 0]

28 Jul 2022

Responses to the comments

The author thanks the editor and the reviewers for their professional and valuable suggestions and has revised the manuscript according to these suggestions, which makes the paper clearer and more readable than the original version. For each suggestion or comment, the detailed answer is given as follows:

Journal Requirements:

Q1: Please ensure that your manuscript meets PLOS ONE's style requirements, including those for file naming. The PLOS ONE style templates can be found at 

A1:We have ensured that our manuscripts comply with PLOS ONE's style requirements, including file naming requirements.

Q2: We note that the grant information you provided in the ‘Funding Information’ and ‘Financial Disclosure’ sections do not match. 

A2:We have checked and corrected the relevant information in the "Funding Information" and "Financial Disclosure" sections in the submission system.

Q3:In your Data Availability statement, you have not specified where the minimal data set underlying the results described in your manuscript can be found. PLOS defines a study's minimal data set as the underlying data used to reach the conclusions drawn in the manuscript and any additional data required to replicate the reported study findings in their entirety. All PLOS journals require that the minimal data set be made fully available. For more information about our data policy, please see http://journals.plos.org/plosone/s/data-availability.

A3:We uploaded the minimal dataset to the submission system as "Figure".

Q4: Please ensure that you refer to Figure 1 in your text as, if accepted, production will need this reference to link the reader to the figure.

A4:We have corrected Fig. 1, please see the supporting information file.

Q5:Please review your reference list to ensure that it is complete and correct. If you have cited papers that have been retracted, please include the rationale for doing so in the manuscript text, or remove these references and replace them with relevant current references. Any changes to the reference list should be mentioned in the rebuttal letter that accompanies your revised manuscript. If you need to cite a retracted article, indicate the article’s retracted status in the References list and also include a citation and full reference for the retraction notice.

A5:We rechecked the references and made corrections.

For Reviewer #1

Q1: Section Basic concepts is not clear enough. Please re-write this section.

A1: The basic concepts covered in this paper have been re-examined and corrected, such as the concepts in line 192.

 Q2: The application of the new proposed model needs to be supplemented.

A2: The application of the newly proposed model has been theoretically supplemented in the research motivation section (lines 163-176), and the specific application of the model is the content of the author's follow-up research.

Q3: There are too many abbreviations in the full text, which is not conducive to readers' reading. It is recommended to optimize. Some abbreviations in the first time don't explain the meaning.

A3: We have supplemented the abbreviations in the full text, such as lines 11, 33, 86, 94, 100, 103, 114, 130, 134, 136, 150. 

Q4: Fig 1 is not good, please change it.

A4: We have corrected Fig. 1, please see the supporting information file.

Q5: Some equations haven’t number. Line 268, 283, etc. It is recommended to identify the equations through numbering.

A5: We have rechecked and renumbered the equations. See lines 223, 224, 260, 261, 264, 287, 291, 294, 320. 

Q6: The quality of Figs. 2-5 needs to improve.

A6: We have made improvements to Fig. 2-5. See supporting information file.

Q7: Conclusions should be more forceful and elaborated.

A7: We have supplemented the conclusion section, see the conclusion section (lines 447-461).

For Reviewer #2

Q1: English needs revision by a native speaker.

A1: We have edited the language on the "editage" recommended by the PLOS ONE journal before submitting the manuscript, and we have rechecked the English grammar.

 Q2: The pros and cons of related work should be summarized in the introduction section, in order to highlight your work's novelty.

A2:Research motivations in the introduction section have been rewritten, see lines 163-176.

Q3: The real problem is not clearly identified. Authors are required to critically analyze the limitations and difficulties in previous studies, add some discussion on the existing methods and highlight what are their strengths and weaknesses so that justifies the development of a new model.

A3:Research motivations in the introduction section have been rewritten, see lines 163-176.

Q4: The materials for the literature review are not very well arranged to justify the proposed solution.

A4:We have re-examined and re-corrected in the literature review section.

Q5:Results and discussions do not clearly support the claimed contribution. Conclusion has to be improved.

A5: The contribution of this paper mainly lies in the improvement of the optimal solution (see Table 1-3), and the accuracy statistics and CPU time of the algorithm (see Table 5-6,8).

Q6:Conclusion has to be improved.

A6: We have supplemented the conclusion section, see the conclusion section (lines 447-461).

---

## [Decision Letter · Decision Letter 1]

11 Aug 2022

Based on an approximation method and related techniques evolutionary algorithm of bilevel programming problems

PONE-D-22-04037R1

Dear Dr. Liu,

We’re pleased to inform you that your manuscript has been judged scientifically suitable for publication and will be formally accepted for publication once it meets all outstanding technical requirements.

Kind regards,

Seyedali Mirjalili

Academic Editor

PLOS ONE

Additional Editor Comments (optional):

Reviewers' comments:

Reviewer's Responses to Questions

**Comments to the Author**

1. If the authors have adequately addressed your comments raised in a previous round of review and you feel that this manuscript is now acceptable for publication, you may indicate that here to bypass the “Comments to the Author” section, enter your conflict of interest statement in the “Confidential to Editor” section, and submit your "Accept" recommendation.

Reviewer #1: All comments have been addressed

Reviewer #2: All comments have been addressed

2. Is the manuscript technically sound, and do the data support the conclusions?

Reviewer #1: Yes

Reviewer #2: Yes

3. Has the statistical analysis been performed appropriately and rigorously? 

Reviewer #1: Yes

Reviewer #2: Yes

4. Have the authors made all data underlying the findings in their manuscript fully available?

Reviewer #1: Yes

Reviewer #2: Yes

5. Is the manuscript presented in an intelligible fashion and written in standard English?

Reviewer #1: Yes

Reviewer #2: Yes

6. Review Comments to the Author

Reviewer #1: (No Response)

Reviewer #2: (No Response)

7. PLOS authors have the option to publish the peer review history of their article (what does this mean?). If published, this will include your full peer review and any attached files.

Reviewer #1: No

Reviewer #2: **Yes: **Nima Khodadadi

---

## [Editor Report · Acceptance letter]

19 Aug 2022

PONE-D-22-04037R1 

An evolutionary algorithm based on approximation method and related techniques for solving bilevel programming problems 

Dear Dr. Liu:

I'm pleased to inform you that your manuscript has been deemed suitable for publication in PLOS ONE. Congratulations! Your manuscript is now with our production department. 

Kind regards, 

on behalf of

Prof. Seyedali Mirjalili 

Academic Editor

PLOS ONE